# Misalignment Between Vision-Language Representations in Vision-Language Models

**Yonatan Gideoni**[†]   **Yoav Gelberg**[†]   **Tim G. J. Rudner**[*]   **Yarin Gal**[†]
[†]OATML, University of Oxford    [*]University of Toronto

## Abstract

Vision-language models fail at some tasks that are simple for humans, but why? Many failures are hypothesized to stem from difficulties translating between the model's vision and language representations. We demonstrate this with a new failure: a VLM can misclassify simple concepts, like a zebra, if they are not explicitly used to align its vision-language representations. This happens even though the vision and language models separately know about the concept, with the failure occurring for both retrieval and generative models. Alignment is difficult as the VLM's language representations cluster based not on semantics but on other features, e.g. the first word in a sentence. We propose a method that can match images and captions without directly translating between their representations and demonstrate that it achieves good performance on a benchmark where VLMs struggle due to representational misalignment, beating models with two orders of magnitude more parameters.

## 1 Introduction

Although large language models (LLMs) are superhuman on many tasks [Achiam et al., 2023], their multimodal counterparts exhibit some very simple failures. Vision-language models (VLMs) struggle in identifying visual analogies [Yiu et al., 2024], matching semantically different but syntactically similar captions to images [Thrush et al., 2022], and counting atypical objects, e.g. the number of legs on a three-legged chicken [Vo et al., 2025].

Many of these stem from differences in the vision and language models' representations. Diwan et al. [2022] ask why VLMs struggle on a specific seemingly simple multimodal benchmark, Winoground, and find that it is likely due to representational misalignment. Coming from a different perspective, Vo et al. [2025] show that VLMs can default to relying on the language model's knowledge instead of on what the vision model actually sees.

It is difficult quantifying what aligned representations are, where solely interpretability-based analyses can be subjective [Lipton, 2018]. An alternative is to test alignment functionally, but what's a good test? One option comes from a thought experiment known as Molyneux's problem — "could a blind man who suddenly regains their eyesight visually recognize objects they previously only felt?" [Degenaar et al., 2024]. Recent research showed that people who regained their eyesight can visually distinguish between objects they feel a few days after their surgery [Held et al., 2011]. Replacing tactile knowledge with linguistic, can VLMs do the same? For example, given a vision and language model that separately know what a zebra is, when combined as a VLM can this new joint model recognize zebras?

Surprisingly, it can't. Typically unimodal backbones are turned into a VLM by training a small vision-to-language adapter that projects vision embeddings into language token representations. To test if the model generalizes between modalities, a concept, like a zebra, can be excluded from its

39th Conference on Neural Information Processing Systems (NeurIPS 2025) Workshop: Unifying Representations in Neural Models.

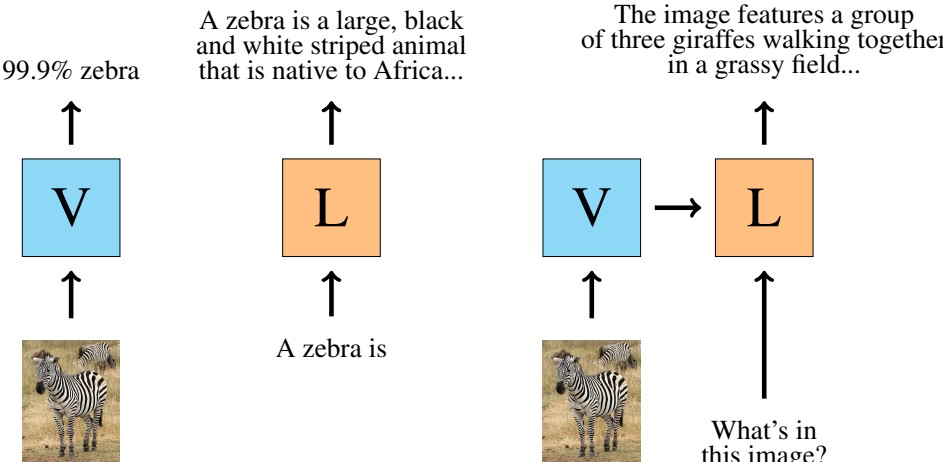

Figure 1: Pretrained vision (V) and language (L) models that both know what a zebra is but still fail to identify one if not jointly trained on zebra image-text pairs. Image from Commons [2024].

multimodal training set. Although the vision and language models separately recognize zebras, the resulting VLM fails at classifying them (see Figure 1).

Here we use this failure as a case study in why VLMs, as they are often trained, struggle in bridging between different representations. To demonstrate a solution we develop Partially Supervised Intermodal Alignment (PSIA), an intermodal retrieval method that relies only on single-modality models. Our contributions are

- Demonstrating a new failure mode of VLMs that functionally tests representational alignment between modalities.

- This failure begs the question, why is representational alignment evidently harder between modalities than within them? We find this is due to language models' representations having a nonlinear structure with respect to some non-semantic variations. For example, representations for "a fox is good" minus "a fox is bad" are similar to "a wolf is good" minus "a wolf is bad" but not to "a wolf is good" minus "*the* wolf is bad". As an image can have several semantically similar captions, this makes the intermodal translation much harder than unimodal translation.

- This leads us to develop an intermodal retrieval method, PSIA, that relies only on unimodal embeddings and intermodal consistency, e.g. where a lion is similar to a giraffe in most modalities. On a difficult representational alignment benchmark, Winoground, PSIA outperforms some 10B+ VLMs while using backbones with fewer than 100M parameters and no gradient-based training.

## 2 Failure Setup

We briefly describe when and how the failure occurs, with a thorough overview relegated to Appendix A. To test whether the unimodal backbones' knowledge generalizes between modalities, we pick several common unambiguous objects to omit from the multimodal training set as held-out concepts.[1] The models are then finetuned on either their regular datasets or a version without the held-out concepts. The models trained without the concepts struggle recognizing them, getting concept detection accuracies around 4-15%, whereas those trained on the regular data get accuracies around 90%. This is shown for both small-scale retrieval and generative settings, using a ViT-B/16 and

---

[1]Specifically, "zebra", "banana", "pizza", "umbrella", and "toothbrush". These concepts were chosen as they appear often in the datasets, are simple, are synonymless, are verifiably known by the models (e.g. all are Imagenet classes, so an image classifier clearly recognizes them), and are easily describable by analogy. For example, "a horse with black and white stripes" is a good approximation of a zebra.

GPT2-small, and using a large-scale generative VLM, LLaVA-7B [Liu et al., 2023]. In the small cases, only linear adapters/projectors are trained so the pretrained models' weights are not altered, with stronger adapters ablated in Appendix B. LLaVA uses low-rank (LoRA) adapters.

# 3    Why does the Failure Happen?

The failure is surprising precisely because we would expect it *not* to occur. Examining why this intuition breaks down helps clarify its causes.

Language representations are known to exhibit analogies. Mikolov et al. [2013a] first showed that the representations of "King" minus "Queen" are similar to "Man" minus "Woman". Similarly, for vision, Radford [2015] showed that this "representation arithmetic" also happens in a GAN, where the representation of "smiling woman" minus "neutral woman" plus "neutral man" yields a smiling man's image. Assuming this holds more generally and for other models, as some works show [Millidge, 2023, Huh et al., 2024], this implies these representations have a linear structure with respect to specific concepts.

Given two models with such a linear structure, a linear transformation should be sufficient to map between their representations.[2] Mikolov et al. [2013b] found that a linear map between different languages' embeddings allows translating words not in that map's training data, with top-1 accuracies over 40%. Why are the accuracies far lower when the models are from different modalities?

To understand why the multimodal case is different we focus on the language model's representations, plotting its representations for various sentences. We use all combinations of "the/a fox/wolf is bad/good/happy/sad" as inputs as they have syntactic and semantic variations with clear similarities between versions. Output representations are taken by averaging the token sequence's representations before the unembedding layer. To generate input embeddings analogous to the generative case, for each sentence 20 single-token soft prompts that generate it are learned. In both cases the language model is GPT2-small.

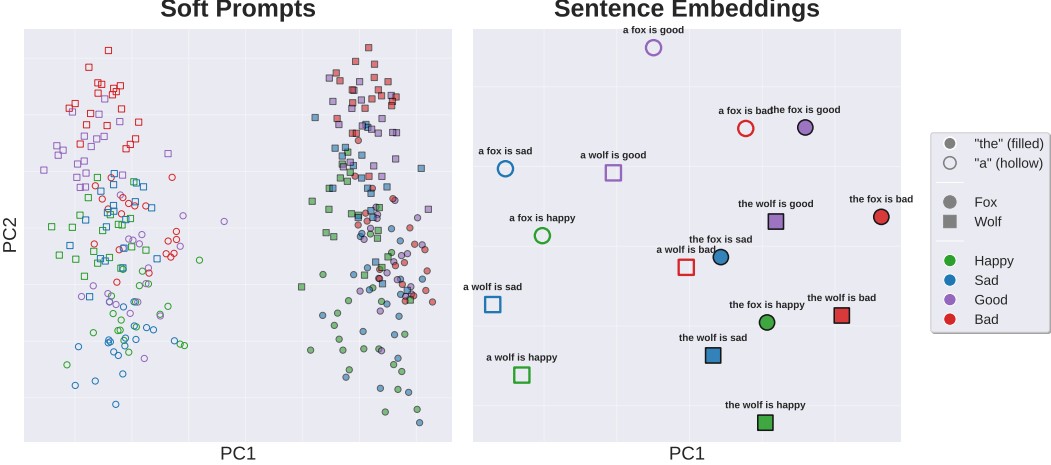

Figure 2: PCA representations of different sentences, at the language model's input (soft prompts that induce a sentence – left) or output (the embeddings it gives for different sentences – right). While some analogies are seen, embeddings cluster chiefly based on the first token ("a" vs "the"), which is a more syntactic than semantic variation. Details are in Appendix G.

Figure 2 shows a PCA of these representations. In both cases there is a clear divide based on the sentences' first words. While some analogical relationships exist, e.g. see Figure 2 right, semantically similar sentences can have far-apart representations. These syntactic variations can lead to complex relationships that make the intermodal mapping nonlinear, e.g. as "a fox is happy" minus "a fox is

---

[2]If $v_k, v_q$ are respectively the representations of king and queen in some modality, and $T$ is a map from one modality to another, then the linear structure means that $T(v_k) - T(v_q)$ should map to $T(v_k - v_q)$ as the same analogies exist in the different representation spaces.

good" is similar to "a wolf is happy" minus "a wolf is good" but the relationship breaks if the last sentence is replaced by a semantically similar phrase like "the wolf is good". The analogies hold only under simple semantic variations.

Thus, given an image of a smiling fox, which caption should it map to? An image can have several semantically similar captions, but each could have a very different representation in the language model. This makes the intermodal mapping more difficult than the intramodal setting.

A different perspective is due to forgetting. If the entire model is finetuned then it is not surprising that it would forget concepts not in the new data. When tuning fewer parameters this could still happen but less so, and similarly when indirectly tuning the model using adapters.[3] In these cases there is tension between the adapter's complexity, the model's final performance, and the incurred forgetting. If the relationship between different representations is simple, as it is in Mikolov et al. [2013a], less forgetting occurs as a simple adapter can easily generalize.

## 4   Partially Supervised Intermodal Alignment

The intermodal alignment problem is one of generalization, and there are several ways to train a model so it generalizes better. One is to have a training signal that is informative about unseen concepts or classes, as happens in knowledge distillation [Hinton et al., 2015]. Another is to better use existing models' capabilities, so learning becomes unnecessary. These considerations motivate PSIA, which at a high level relies on unimodal embedding models and consistency between modalities, without needing any direct representational translation.

Knowledge distillation cannot be directly used for retrieval as it is designed for classification. However, many retrieval settings construct semantically meaningful pseudo-classes – given a set of anchors $A$ and candidates $C$, a semantic similarity model $s$ can assign each candidate $c_i$ a probability vector $s(c_i) = p_i$ over the anchors.[4] Two well-trained semantic similarity models $s_a, s_b$ should be consistent with one another, so $s_a(c) \approx s_b(c)$. Additionally, given two semantically identical versions of the same data (e.g. views of an image or descriptions in different modalities), $c, \tilde{c}$, a well-trained model should be consistent here too, having $s(c) \approx s(\tilde{c})$.

This consistency is powerful because it describes an inherent relationship between data in different modalities. Specifically, for a vision model $s_v$ and text embedding model $s_t$, given an image-caption pair $v, t$, consistency requires that $s_v(v) \approx s_t(t)$, where here the anchors are image-caption pairs. Note that this connection does not require any additional gradient-based training over the paired data.

In practice, when doing retrieval, the candidates can be seen as two sets, one with images and one with captions, instead of a single set of pre-existing pairs. The consistency requirement prescribes how to do intermodal retrieval, such as text-to-image lookup. Given some text $t$ and a set of candidate images $v_1, v_2, ...,$ find $\arg\min_i d(s_t(t), s_v(v_i))$, where $d$ is a distance metric between the two distributions. We use the Jensen-Shannon divergence as our distance of choice as it works well in practice.[5] Additional technical details are in Appendix H.

We demonstrate PSIA over Winoground, a benchmark where models need to distinguish between pairs of similarly worded captions

| Model | Winoground Accuracies | | | Total # |
| | Text | Image | Group | Parameters |
| --- | --- | --- | --- | --- |
| CLIP (ViT-B/32)* | 30.8 | 10.5 | 8.0 | 151M |
| LLaVA-1.5-13B† | 33.5 | 35.0 | 17.3 | 13B |
| GPT4V† | 60.3 | 45.3 | 33.5 | >1T |
| PSIA | 46.8 | 43.8 | 42.5 | 120M |
| Random* | 25.0 | 25.0 | 16.7 | |
| Human* | 89.5 | 88.5 | 85.5 | |

Table 1: Winoground scores for different models. Text, image, and group refer to text→image, image→text, and joint matching accuracies, respectively. Random chance is lower than 50% because a set of two images and two captions is correctly labeled only if both pairs are correctly matched. Experiment details are in Appendix H.1. *Thrush et al. [2022],†Mitra et al. [2024]

and their images, e.g. "some plants surrounding a lightbulb" and "a lightbulb surrounding some plants" [Thrush et al., 2022]. Table 1 shows that PSIA works well, surpassing LLaVA-13B. Notably, the

---

[3]This is intuitive but nontrivial, see Appendix C for a deeper discussion.

[4]For brevity, throughout this section conditioning is omitted and we use $s(c_i)$ for $s(c_i|A)$.

[5]Technically Jensen-Shannon is a distance squared, but squaring does not affect relative rankings between possible candidates. Also, as this is a comparison between two distributions with one acting as a reference, it is tempting to use divergences like Kullback-Leibler instead of distances, but this yielded worse performance.

anchors here are random image-caption pairs from the COCO dataset, which is a qualitatively different distribution than Winoground. PSIA gets a better group accuracy than GPT4V and outperforms LLaVA-1.5-13B on all metrics, in spite of it not requiring any gradient-based training over multimodal data and using far smaller models.

# 5 Discussion

Vision and language representations are likely difficult to align due to having fundamentally different structures. Language's sequential order makes semantically similar sentences have very different embeddings, unlike vision where models learn to cluster semantically similar images [Chen et al., 2020]. While these oddities can be consistent within modalities, evidenced by simple in-modality alignment between models as shown by Mikolov et al. [2013a], they can lead to surprising shortcomings when the modalities are different.

It is unclear how scale affects this failure. While seemingly fundamental to how models are currently trained, with the language part of the VLM pretrained only on text, Huh et al. [2024] theorize that non-semantic differences in representations might decrease with model size. Representational alignment is known to be difficult even with very large models, as they still have sub-human performance on some simple multimodal tasks [Thrush et al., 2022, Vo et al., 2025, Yiu et al., 2024]. On the other hand, Geirhos et al. [2021] report that, in some cases, larger vision models gradually make more "human-like" mistakes — perhaps VLMs do as well.

PSIA demonstrates that the two modalities can be bridged without explicitly projecting from one representation into another. As it is only for retrieval, it cannot be used for many problems that require generation, but it shows a conceptual potential solution. Finding ways to generalize and robustify PSIA is an interesting avenue for future work.

What may fix this failure for generative models? Using a language-aligned vision backbone, like a CLIP model, would alleviate but not fully solve it as the language model's embeddings would remain the same. It would be interesting to see if large VLMs trained without different language and vision components, such as Fuyu-8B [Bavishi et al., 2023], exhibit these failures, but their pretraining is costly. Finding efficient ways to create such models, e.g. by merging unimodal components, could be interesting for future work.

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

## A   Failure Demonstrations

VLMs are typically trained by combining pretrained vision and language models and finetuning them on a dataset of image-text pairs. For example, for image captioning an adapter can be used to project the vision model's representations past the language model's embeddings layer as a "soft prompt" that induces the correct caption [Merullo et al., 2022]. An extended discussion of common VLM setups is given in Appendix E.

**Datasets.** To test generalization we remove the held-out concepts from the finetuning data. This is done by filtering out all image-text pairs where a) the text contains the object's name or b) the image is likely to contain it. Image-based filtering is important to prevent misclassified data from hampering the experiment, e.g. zebra images accidentally captioned as giraffes. This is done using a CLIP ViT-L/14 model and filtering images with a similarity threshold greater than 0.2 to the text "a photo of a <concept>", with the threshold found through manual tuning.

To see how well the models later detect these concepts we construct a test-set consisting of clear examples of them. Using the same CLIP model, for each concept we take the top-100 Imagenet images that are most similar to it.

**How do we know the models "know" these concepts?** For the vision models this is simple – all the concepts are Imagenet classes and the models are Imagenet classifiers that get high accuracies. For the language models, they can be prompted to complete texts such as "a <concept> is" or asked questions about it.

**Experiment details.** We demonstrate the failure for two settings — retrieval and generation. For retrieval we combine ViT-B/16 and GPT2-small models by linearly projecting their final layer's representations to a shared space and training using a CLIP loss [Radford et al., 2021]. For generation we combine the same models but using a single linear adapter, from the vision representations to the language embeddings, training the model to do image-captioning. Both cases use the COCO-2014 dataset for training [Lin et al., 2014] and train only the projectors/adapters, keeping the base vision and language models fixed. To ensure the models are well trained their learning rates and weight decays are found using a random hyperparameter search. Training details and ablations over more complex adapters are in Appendix E.

To see whether the failure disappears at larger scales we also train a LLaVA-7B model [Liu et al., 2023]. LLaVA is a generative VLM trained to do visual question answering using a large curated dataset.[6]

## A.1 Failing to Retrieve

To see whether the VLMs manage to bridge between different modalities we train two models, one trained on the regular dataset and one without the held-out concepts. To measure how well the models retrieve the held-out concepts, we mix the concept test images with some from Imagenet, taking an image from every class that does not include the concepts. Imagenet is a nice "distractor" dataset as it has object-centric images that are easy to filter based on their class labels. To retrieve these we measure the cosine

| Model | Best Concept Detection Accuracy Throughout Training |
|---|---|
| No held-out concepts | 4.5% |
| Regular | 87.4% |
| CLIP (ViT-B/16) | 84.6% |

Table 2: Best concept top-1 detection accuracy over the test set when evaluating throughout training models trained on data with and without the concepts. The model trained without the concepts fails to generalize to them. The regular model performs close to a pretrained CLIP with a similar number of parameters, showing the failure is not due to insufficient model capacity or training. The CLIP model is evaluated as is without additional training.

similarity between the image embeddings and the text embedding of "a photo of a `<concept>`", picking the image with the highest similarity.

Table 2 shows the best concept detection accuracy throughout training using the regular dataset and the one without the concepts. The model trained without the concepts performs poorly, about $20\times$ worse than the model trained on the regular dataset but significantly above random chance (0.1%).

## A.2 Failing to Generate

For the generative setting we test whether the models translate between the modalities by similarly training two models, once on the regular dataset and again without the held-out concepts. A model detects a concept if it directly appears in its generated textual output. Qualitative explanations, like calling a banana "a yellow fruit", hence do not count.

| Model | Best Concept Detection Accuracy Throughout Training |
|---|---|
| No held-out concepts | 7.8% |
| Regular | 89.6% |
| End-to-end finetuning[†] | 91.3% |

Table 3: Best concept detection accuracy throughout training. The model trained on data without the held-out concepts does far worse than the one trained with them. [†][NLP Connect, 2022]

For the small-scale image captioning task Table 3 shows the top concept detection accuracy throughout training. The model trained without the held-out concepts fails to generalize, while the model trained on the regular dataset performs similarly to a strong baseline from NLP Connect [2022] using the same unimodal backbones, showing that training just the linear adapter does not substantially hurt performance.

---

[6]We do not test how scale affects retrieval as large-scale multimodal retrieval tasks usually use pretrained CLIP models, unlike generation where powerful unimodal models are often finetuned. Thus, we are unaware of cases where large pretrained unimodal models are combined and finetuned for retrieval.

### A.2.1 Failing at Scale

Many phenomena are qualitatively different, if not nonexistent, when using sufficiently large models and datasets. To check whether this failure vanishes in a larger setting, we perform a similar experiment with LLaVA-7B [Liu et al., 2023]. Typically LLaVA is trained in two steps, first by finetuning a linear adapter between a frozen vision backbone and language head, and then by finetuning the adapter and language model together. In the second stage, we use low-rank (LoRA) adapters instead of fully fine-tuning the model, both for simplicity and to reduce catastrophic forgetting of the held-out concepts in the language model.

By default LLaVA uses a CLIP-based vision transformer, but this model has already seen concepts in both modalities. Instead, we use a ViT-L/16@384 model that is trained for Imagenet classification. This backbone only saw images during its pretraining and has the same number of parameters and tokens as the regular LLaVA vision backbone.

To test if the model recognizes a concept we prompt it to answer "What is in this image?", with a correct answer being one where it names the concept.[7]

As Table 4 shows, the models trained on the filtered data struggle to generalize. However, they perform better than they do at smaller scales, especially when only the linear

| Model | Concept Detection Acc. (End of Training) |
|---|---|
| No held-out concepts (Stage 1, only linear adapter) | 19.7% |
| No held-out concepts (Stage 2, adapter+LoRA) | 11.7% |
| Regular | 93.5% |
| CLIP Backbone | 98.0% |

Table 4: Concept detection accuracy for different models. Accuracy is shown at the end of training to emphasize the different stages for the model trained without the concepts. "Regular" is after stage 2 using the ViT backbone and the unfiltered dataset, as is "CLIP Backbone" but with LLaVA's default vision transformer.

adapter is used. The best no held-out concepts detection accuracy throughout the first stage of training peaks at 31%, which implies the failure may have the potential to disappear at scale but be nullified by the continued training. However, it is unclear to what extent the model properly recognizes objects vs memorizes associations, e.g. that bathrooms typically have toothbrushes — this is discussed and qualitatively analyzed in Appendix F.

## B  Do Better Adapters Help?

To see if better adapters help we repeated the two small-scale experiments with two more powerful adapters – an MLP and an attention block between the tokens and learnable embeddings. The attention adapter has a single learnable token embedding for retrieval and 200 for generation, where a ViT-B/16 has 197 vision tokens by default. For retrieval we also experiment with three different kinds of pooling – mean pooling, max pooling, and, specifically for the vision transformer, taking the class token. When the vision transformer uses the class token the language model defaults to mean pooling. The attention adapter needs no pooling due to the cross attention already aggregating information.

In all cases, the linear adapter generalized best to the held-out concepts. For generation the best concept detection accuracy is the one reported in Table 3, whereas for retrieval the pooling with a linear adapter did a bit better, getting a best concept detection accuracy throughout training of 11.1%.

Note that this is a very lenient metric as it measures the model's best performance at any point throughout training, so it is unaffected by overfitting. In practice, test accuracies were measured after every 0.1 epochs.

## C  Why Could Adapters Lead to Forgetting?

There are a few explanations for why tuning only part of a model, or layers added to the start or end of it, can result in forgetting. Broadly, given sufficient optimization and strong adapters, this can be

---

[7]Other prompts, such as trying to promote reasoning by asking it to "Describe the image's features without naming any objects and afterwards please describe the image's contents", were found to worsen performance.

equivalent to training the model as usual. For example, Petrov et al. [2024] show that training soft prompts is in some cases equivalent to finetuning the full model.

Specifically for models that output softmax distributions (like generative language models) the optimal distribution during training for a never seen token is one where it gets zero probability and hence a very negative logit. A simple way to do this is by making its logit bias very low, which a linear layer on top of the model's head can easily do.

In all these cases there is no guarantee the model will forget and not benignly overfit to the data while still generalizing. Whether this happens likely depends on how powerful the adapter is and how complex the relationship it needs to learn.

## D   Hyperparameters

Optimal learning rates and weight decays were chosen by doing a grid search over learning rates and weight decays spaced log-linearly in $10^{-5}, 10^{-4.5}, ...10^{-2}$ and $10^{-4}, 10^{-3}, ...10^{-1}$ respectively. Optimal retrieval hyperparameters were chosen based on the model's top-1 Imagenet classification accuracy over Imagenet's validation set. Generation hyperparameters were optimized on the BLEU score over COCO's validation set. Models were trained for 10 epochs with an early stopping patience of 5, with validation metrics computed once every 0.25 epochs.

Resulting optimal hyperparameters for all generative model configurations are listed in Table 5. For retrieval a learning rate of $10^{-4}$ and weight decay of $10^{-2}$ was found to work well for all settings.

| Adapter type | Learning rate | Weight decay |
|---|---|---|
| Linear | $10^{-2.5}$ | $10^{-2}$ |
| MLP | $10^{-3}$ | $10^{-2}$ |
| Attention | $10^{-3.5}$ | $10^{-3}$ |

Table 5: Optimal learning rates and weight decays for different adapters for the small-scale generative experiments.

For the large-scale experiment with LLaVA the default hyperparameters and training setup as per Liu et al. [2023] were used.

## E   How are VLMs Trained?

We focus first on retrieval VLMs, which are typically CLIP models [Radford et al., 2021], and then on generative VLMs, where we focus on models that output text.

Training is usually as per Radford et al. [2021], where a text and vision encoder are trained together from scratch using image-caption pairs. CLIP outputs embeddings where their temperature-scaled cosine similarities are interpreted as logits. CLIP is trained to maximize the probability between correct image-caption pairs and minimize incorrect pairs' matching probabilities using a cross entropy loss.

Generative VLM training is more involved. Models are often initialized using pretrained vision and language models, with them being connected using some adapters. While many variants exist [Alayrac et al., 2022, Li et al., 2023, Liu et al., 2023], Merullo et al. [2022] showed that a linear adapter is sufficient. This adapter projects vision tokens from the vision transformer into the language model, acting as "soft prompts" [Li and Liang, 2021] – token embeddings that result from some upstream representation instead of the model's embedding layer. In many cases the adapter and language model are trained together, although Zhu et al. [2023] find training just the adapter to be sufficient.

Often the vision backbone is a CLIP model's vision encoder so its representations are already somewhat aligned with the language model due to its multimodal training. Although not studied here, this can lead to some failures as well, e.g. Gal [2021], Tong et al. [2024] find that CLIP models can give similar representations to images which have meaningful semantic differences.

## F  Recognizing vs Memorizing in LLaVA

A VLM can coincidentally say that a concept is in an image without recognizing it but hallucinating it based on its surroundings. Vo et al. [2025] explicitly show this bias by demonstrating VLMs misclassifying how many legs edited pictures of animals have, where the VLM's language part relies on pre-existing knowledge (e.g. "a chicken has two legs") instead of using the picture ("this chicken has three legs"). In our case such hallucinations are more subtle, as it is unclear whether the VLM recognized a concept visually or relied on the language model's knowledge. For example, given an image of a bathroom the VLM may start outputting the sentence "This is an image of a bathroom, which has a soap dispenser, toothpaste, ..." and at some point mention a toothbrush, regardless of whether or not the image has one.

It is hard to tell when the VLM recognizes a concept versus relies on its surroundings and the language model's knowledge. Manually checking the answer LLaVA gives after its first stage of training (top row in Table 4) shows that this may be about 30% of answers where the concept was detected. An answer is considered associative and not direct if before the model mentions the main concept (e.g. a zebra) it mentions its surroundings or characteristics in a way where the language model would likely have mentioned the concept anyway, e.g. the bathroom example from before.

## G  Language Representations Plot Details

The sentence embedding/"output representations" are found by passing the sentence through GPT2-small and averaging the representations before the unembedding layer. The soft prompts/"input representations" are found by optimizing a soft prompt made up of a single token to produce a given sentence. Each soft prompt was randomly initialized from a standard normal distribution and optimized for 200 steps with a learning rate of 0.1. All final soft prompts perfectly reproduce their target sentence.

## H  Additional Details on PSIA

PSIA has some implicit assumptions which are important to address. When using different semantic similarity models $s_t, s_v$ the assumption that $s_t(t_i) \approx s_v(v_i)$ requires both models to be equally calibrated with respect to their temperature. For example, if the text model $s_t$ has a very high temperature then it will give a uniform distribution across anchors and it will not be consistent with the vision model's distribution. This can be remedied by calibrating the two temperatures using a validation set, or manually finding a good temperature.

Another problem is whether the two modalities contain the same information. The captions are generally less informative than the pictures, as to fully specify an image requires a very long description. While this can lead to wrong candidate pairings, given a sufficiently large pool of candidates all matches should be at least semantically plausible.

### H.1  PSIA Experiment Details

The experiment in Section 4 used a sentence transformers all-MiniLM-L12-v2 [Reimers and Gurevych, 2019] and MoCo v3 [Chen et al., 2021] models for the language and vision embeddings respectively. MoCo v3 was chosen as it is an instance discrimination model [Wu et al., 2018], trained to output probabilities between pairs of embeddings, although ongoing experiments show that other kinds of models can work as well. all-MiniLM-L12-v2 has 33M parameters while MoCo v3 is based on a ViT-B/16, which has 86M. For simplicity a temperature of 10 was used for both modalities.

## I  Compute

All small-scale experiments were run on a single A100 GPU with each run taking a few hours. LLaVA-7B was trained on a single GH200 over a bit more than a day. PSIA was run on an RTX 4090 and takes a few minutes, including loading and sampling anchors.

