# OpenReview forum: "Misalignment Between Vision-Language Representations in Vision-Language Models"
_NeurIPS.cc/2025/Workshop/UniReps — UniReps2025_

### Official Review · Reviewer_NVRQ · 2025-09-11
**Review of Misalignment Between Vision-Language Representations in Vision-Language Models**

**Confidence:** 3

**Review:**

### Summary
The authors attribute simple failures by VLMs to misalignment between the vision and the language models' representations. To validate their claim, the authors align these models by excluding some "concepts" that both the models have seen before. The resulting VLM fails to identify the excluded concepts, with accuracies falling significantly from their counterparts trained on the complete dataset. Finally, the authors propose Partially Supervised Intermodal Alignment (PSIA) for cross-modal retrieval. Given a set of Anchors A (image, text) pairs, a similarity model can assign a probability vector **s(i)/s(c)** for a candidate **i/c** (image/caption). The authors use the prior that for an image *i* and its caption *c*, s(i) and s(c) should be consistent. Hence, PSIA can be used for retrieval by minimizing the distance between these distributions without explicitly aligning vision-similarity model and language-similarity model. PSIA outperforms CLIP, LLaVA and GPT4V without requiring any training and being much smaller in size.

### Strengths
- The failure in alignment of held-out concepts has been well explained and clearly demonstrates the significant gap between models trained with and without the concept.
- The proposed method PSIA outperforms LLaVA and GPT4V despite being significantly smaller in size without explicit alignment.
- The overall flow of the paper is good.
### Weaknesses
- Evaluation: The method has only been evaluated and compared on a single benchmark. Also, the evaluation could benefit from including more open-source models such as Qwen VL and InternVL.
- As mentioned in the paper, the proposed method is limited to retrieval.
- Minor: PSIA's setup in Section 4 is difficult to read.

### Suggestions
- The paper should discuss PSIA's robustness to different Anchor sets.

**Score:**

4

**Topic Fit:**

3

---

### Official Review · Reviewer_Kgzo · 2025-09-15
**Misalignment in Vision-Language Models Representations**

**Confidence:** 4

**Review:**

# Summary

The authors demonstrate a striking failure case: even when both unimodal components (vision and language models) individually know a concept like “zebra,” the combined VLM can fail to recognize it if the concept was not explicitly present in multimodal training. The work attributes this failure to structural differences in the representational spaces of language and vision, where language embeddings cluster by syntactic rather than semantic properties. To address this, the authors propose PSIA, a retrieval-only method that leverages consistency between semantic similarity models across modalities.

# Strengths

- The PCA analysis showing that language embeddings cluster by syntactic tokens (e.g., “the” vs. “a”) is a compelling illustration.
- On Winoground, PSIA outperforms LLaVA-13B and approaches GPT-4V in group accuracy, despite being orders of magnitude smaller.

# Weaknesses

- Limited Scope of Held-Out Concepts: The study uses a handful of objects (“zebra,” “banana,” etc.). While chosen carefully, broader concept classes (verbs, abstract entities) would test the generality of the phenomenon.
- Dataset scope is too narrow: Evaluation is restricted to Winoground, which is among the smallest relational benchmarks and may not generalize. For stronger evidence, the method should also be tested on SugarCrepe, VG Relation/Attribution, or BiVLC, which are richer in compositional and relational reasoning.
- Choice of language model for analysis: The PCA study uses GPT-2 small, which is not the LLM backbone of LLaVA. To convincingly argue misalignment, it would be better to analyze representations from the actual VLM backbone (or at least a larger, comparable LLM).

**Score:**

3

**Topic Fit:**

3

---

### Official Review · Reviewer_A36J · 2025-09-16
**VLM representation misalignement and how to fix it**

**Confidence:** 4

**Review:**

There are differences between language model representations and vision model representations. Those impact the quality of mixed models, which in some cases are misaligned. This work first looks at this by retraining vision-language adapters with some data removed, and shows multimodal systems fail to generalize to unseen data. They then propose a method to align multi-modal representations which requires little data and works very well on preliminary benchmarks and models.

For the first part about semantic misalignement, further evidence (in future works, depending on the direction you choose for this) could include more extensive evaluation of semantic space. Qualitative methods include Sparse AutoEncoders, or GradientCam. Quantitative methods which are robust to the high dimensional spaces of those models include information imbalance, K nearest neighbours, CKA, CCA,... while PCA is a great visualisation of the distribution of datapoints, it is difficult to use it to prove a negative and say the expected structure is NOT in the representation. Perhaps there exists an organisation of the data that we cannot see with this method.

Hypothesis that are proposed to explain the difference between vision and language (catastrophic forgetting, misalignement, ...) are interesting and definently something to study further. It is my understanding that recent VLM models retrain completely for alignement, and either train a translator between modalities, effectively not harming any modalitie's model weights, or directly train on both modalities. I would think that this rules out catastrophic forgetting for usual methods.

In paragraph 4.2 from the following paper (Maiorca, Valentino, et al. "Latent space translation via semantic alignment." Advances in Neural Information Processing Systems 36 (2023): 55394-55414.) results seem to show direct alignement between modalities, using a very similar anchor based method on a multi-modal setting. Works from these authors extensively study representational alignement through many methods - I would recommend comparing to their methods in future, to clarify novelty of your method, and perhaps show performance gains.

In future works, I think you might be interested in the following (recent, understandably not cited) works:
This one comes up with a very informative setting to see in which modality information used by the model is found:
Ortu, Francesco, et al. "When Seeing Overrides Knowing: Disentangling Knowledge Conflicts in Vision-Language Models." arXiv preprint arXiv:2507.13868 (2025).

In this one there is a quantitative method to look at semantic differences between modalities :
Acevedo, S., Mascaretti, A., Rende, R., Mahaut, M., Baroni, M., & Laio, A. (2025). An approach to identify the most semantically informative deep representations of text and images. arXiv preprint arXiv:2505.17101.

TLDR: I think this is interesting work, I have some doubts on the experimental backing of hypothesis proposed in the first part, and of novelty and performance gains for the second part, but consider that this is work worth considering which could serve the community by strengthening understanding of differences between modality representations.

**Score:**

2

**Topic Fit:**

3